# Promising Potential of *Lonchocarpus utilis* against South American Myasis

**DOI:** 10.3390/plants9010033

**Published:** 2019-12-25

**Authors:** Carmen X. Luzuriaga-Quichimbo, José Blanco-Salas, Carlos E. Cerón-Martínez, Juan Carlos Alías-Gallego, Trinidad Ruiz-Téllez

**Affiliations:** 1Faculty of Health Sciences Eugenio Espejo, University UTE, Quito 170147, Ecuador; luzuriaga.cx@gmail.com; 2Área de Botánica, Facultad de Ciencias, Universidad de Extremadura, Avda. Elvas s/n, 06071 Badajoz, Spain; truiz@unex.es; 3Herbarium Alfredo Paredes, QAP, Ecuador Central University UCE, Quito 170521, Ecuador; carlosceron57@hotmail.com; 4Área de Ecología, Facultad de Ciencias, Universidad de Extremadura, Avda. Elvas s/n, 06071 Badajoz, Spain; jalias@unex.es

**Keywords:** *Lonchocarpus utilis*, barbasco, rotenoid, in silico, drug discovery, bioinformatic

## Abstract

Traditional medicine is especially important in the treatment of neglected tropical diseases because it is the way the majority of populations of affected countries manage primary healthcare. We present a case study that can serve as an example that can be replicated by others in the same situation. It is about the validation of a local remedy for myasis in Amazonian Ecuador, which is contrasted by bibliographic chemical reviews and in silico activity tests. We look for scientific arguments to demonstrate the reason for using extracts of *Lonchocarpus utilis* against south American myasis (tupe). We provide a summary of the isoflavonoids, prenylated flavonoids, chalcones, and stilbenes that justify the action. We make modeling predictions on the affinity of eight chemical components and enzyme targets using Swiss Target Prediction software. We conclude that the effects of this extract can be reasonably attributed to an effect of the parasite that causes the disease, similar to the one produced by synthetic drugs used by conventional medicine (e.g., Ivermectine).

## 1. Introduction

The World Health Organization (WHO) has estimated that more than 80% of the world’s population routinely uses traditional medicine to meet their primary healthcare needs [1], and many traditional treatments involve the use of plant extracts or their active ingredients [2]. This is especially important in the treatment of neglected tropical diseases (NTD) [1]. The WHO has recognized twenty NTD: buruli ulcer, chagas disease, dengue and chikungunya, dracunculiasis (guinea-worm disease), echinococcosis, foodborne trematodiases, human african trypanosomiasis (sleeping sickness), leishmaniasis, leprosy (Hansen’s disease), lymphatic filariasis, mycetoma, chromoblastomycosis and other deep mycoses, onchocerciasis (river blindness), rabies, scabies and other ectoparasites, schistosomiasis, soil-transmitted helminthiases, snakebite envenoming, taeniasis/cysticercosis, trachoma and yaws (endemic treponematoses). These diseases represent an important cause of morbidity, disability, and mortality in the poorest people living in developing countries. They are so named because of the lack of financial investment in the development of new drugs by high-income countries to treat them [3]. In this context, the Amazonian countries have large sectors of the population that use plants from the tropical forests where they live daily as a legacy of their ancestors. This represents knowledge that can be articulated with the Western scientist [4,5]. and may produce advances in the field of health. We present a case study that can serve as an example for future replicate studies in this direction.

We investigate the plant known in Spanish as barbasco or poison rope, *Lonchocarpus utilis* A.C. Ye. (*Lonchocarpus nicou*, *Lonchocarpus nicou* var. *languidus*, *Lonchocarpus nicou* var. *urucu*, *Deguelia utilis*, Fabaceae). This is a wild bush plant found in the rainforests of Brazil, Colombia, Guyana, Guyana, Peru, Surinam, Venezuela, and Ecuador, which is sometimes cultivated in indigenous communities for its use [6]. It has different common names according to the original language of the population (e.g., shili bun (tsafi’ki), anku hanpi, auka hanpi, hanpi, lumu hanpi, shikitu hanpi, timun hanpi, tullu hanpi, waska hanpi (kichwa), avu signo’mba, macoroje’cho indica’mba, seña’mba (a’ingae), airo eó, eó, eopo eó, jo’ya eó (pai coca), kompago, kompagon, konpago, meneko (wao tededo), timiu (shuar chicham)).

In Ecuador, several indigenous nationalities have used it both in traditional medicine and ancestral fishing procedures (Table A1). During a study carried out by one of the authors [7] in the Bobonaza river province of Pastaza, some uses (Table A2) that had rarely been mentioned in the area were observed [8,9,10,11]. These provide good examples of the use of natural resources due to the degree of isolation of the communities.

In this case, the plant was employed to treat a very significant illness locally named the “tupe”. This is a myiasis involving the infestation of dipterous larvae favored by the tropical hot and humid climate. More than 170 million people are at risk of contracting this neglected tropical disease [12]. In South America, the largest number of cases are produced by *Dermatobia hominis*, an autochthonous species that acts as a parasite of living tissues [13,14,15]. When the female is willing to lay eggs, she catches a blood-sucking arthropod, a fly or mosquito (mainly of the genus *Psorophoro*), that acts as an intermediate host and deposits eggs (15 to 30) on her abdomen, which are fixed with a kind of adhesive. The intermediary distributes the eggs when looking for animal or human blood to feed on and these will hatch, releasing young larvae that penetrate the skin over a time period that varies from 5 to 10 min in the location of the bite or through the hair follicles [16]. The preferred places in humans are the trunk, thighs, buttocks, and back [17]. Initially, they are skin lesions of little relevance. Infections are unlikely because the larva itself secretes antibiotics as an adaptive strategy to have food in good condition [18]. Infection is much more likely following scratching, handling without conditions of asepsis, or if the larvae are only partially extracted because remains are left under the skin [19]. In this case, it can turn into erythematous papules that increase in size, becoming pustular. If the larvae penetrate further, they form subcutaneous nodules of 1–2 cm, which can form painful abscesses. There may be regional lymphadenopathy, lymphangitis, and eosinophilia. This can affect the skin, mucous membranes, intestine, genitourinary system, lungs, and brain (migration of larvae by fontanelles) [20]. Others, such as those produced by *Sarcopiiaga tiaemorriioidaiis*, *S. iambens*, *Cynomyopsis cadaverina*, *C. vicina*, *Phaenicia sericata,* and *P. cuprina* are also common if there are predisposing factors such as poor hygiene, advanced age, or poor peripheral vascular circulation [21]. More rare are *Musa domestica, Stomoxys caicitrans,* and *Fannia* spp., which lay eggs on open lesions [22,23,24]. A great number of cases of myasis are associated with vulnerable people living in rural areas and poverty or underdevelopment [12]. Given the importance of this neglected pathology, the present bibliographical study was designed.

In addition to this, molecular docking studies have been revealed as a useful tool to predict activity [25,26,27,28,29] and to orientate pharmacological research, saving time, and economic resources. Arguments and hypotheses can be reinforced or confirmed using in silico tests. These approaches are becoming more frequent. As was recently written in *Nature* [30], “bioinformatics can boost basic science in countries with limited resources”. This can be especially useful for NTD.

Based on the above, our research objective is to find experimental evidence of chemical composition and activity, aimed at a scientific validation of myasis treatment employed by the Kichwa people from Amazonian Ecuador.

## 2. Results

The phytochemical composition of *L. utilis* has been widely studied [31,32,33,34,35,36,37,38]. The most important compounds found in the leaves, stems, and roots are rotenone (44%), rotenolone (6.7%), deguelin (22%), and thephrosin (4.3%) [39], which are rotenoids—isoflavones with additional pyrane/furane rings. Other significant components are prenylated flavanols (prenyl-urucuol A, prenyl-isotirumalin and prenylutilinol), prenylated flavones (3′-methoxylupinifolin), prenylated flavanones (2*S*)-6-(γ,γ-dimethylallyl)-5,4′-dihydroxy-3′-methoxy-6″, 6″-dimethylpyran [2″,3″:7,8] flavanone, and prenylutiline, chalcones (4-hydroxylonchocarpin), and stilbenes (lonchocarpene, methoxylonchocarpene, 3,5-dimethoxy-4-*O*-prenyl-*cis*-stilbene and 3,5-dimethoxy-4-hydroxy-3-prenyl-trans-stilbene). The structures are organized [40] and summarized in Figure 1. 

The biological activity of these components, experimentally tested by different authors, is summarized in Table 1.

Figure A1, Figure A2, Figure A3, Figure A4, Figure A5, Figure A6 and Figure A7 (Appendix A) belong to the Swiss Target prediction report files obtained using the corresponding cited structures as query molecules: Figure A1—rotenone; Figure A2—rotenolone; Figure A3—deguelin; Figure A4—tephrosin; Figure A5—3′metoxylupinifolin; Figure A6—4-hydroxylonchocarpin; Figure A7—lonchocarpene.

As it can be observed in the probability graphics of these figures, rotenoids have shown the most affinity for ornithine decarboxylase (ODC), tyrosyl-DNA phosphodiesterase 1, arginine decarboxylase, NADH ubiquinone oxidoreductase chai-4, and the microtubule-associated protein tau (the latter especially for rotenone and rotenolone). Rotenone has shown the best relations with the cytochrome P450 group of enzymes.

Chalcones (4-hydroxylonchocarpin) have shown the most affinity with ODC, tyrosyl-DNA phosphodiesterase 1, arginine decarboxylase, and have a probability of more than 50% with different protein kinases.

The prenylated flavone that was tested in our in silico experiments (3′-methoxylupinifolin) showed affinity with ornithine decarboxylase (ODC), tyrosyl-DNA phosphodiesterase 1, and arginine decarboxylase. Lonchocarpene (stilbene) showed affinity with ornithine decarboxylase (ODC), tyrosyl-DNA phosphodiesterase 1, and arginine decarboxylase.

Another stilbene that has been tested (3,5-dimetoxy-4-hydroxy-3 prenyl-trans-stilbene) has not shown any reliable affinity nor activity.

## 3. Discussion

The hospital treatment of tupe is surgical: the application of local anaesthetics and removal of the larvae through the entrance orifice. For oral medical treatments, the drugs that might be employed are thiabendazole imidazols and macrocyclic lactones. Thiabendazole inhibits fumarate reductase, an enzyme specific for helminths. It is absorbed rapidly in the gastrointestinal tract, is metabolized in the liver, and it is eliminated by the kidney. However, it has many side effects. Ivermectin [58,59] has more selective activity with few systemic effects on mammals. It acts by binding to the anionic glutamate channels of gamma amino butyric acid (GABA) on the nerves and muscle cells of invertebrates, causing pharyngeal paralysis and death of the parasite by asphyxia and starvation. Our results for the Swiss Target prediction reports of isoflavonic rotenoids such as rotenone and rotenolone revealed strong interactions with many cytochrome P450 isozymes, NADH-ubiquinone oxidoreductase, and the microtubule-associated protein tau. Another rotenoid, deguelin, is especially akin to ornithine and arginine decarboxylases, and tephrosin shows affinity to tyrosyl-DNA phosphodiesterase, due to the probability levels shown in the Appendix A predictions. These results obtained by our in silico tests explain the activities of *L. utilis.* We also found a strong relation to the former decarboxylases and other groups of molecules of the extract: stilbenes (as lonchocarpene), chalcones (as 4-hydroxylonchocarpin), and prenylated flavones (such as 3′methoxylupinifolin). Rotenoids have the capacity to act against multiplication or growth, as summarized in the experimental results of Table 1. They cause a lack of energy, respiratory depression, respiratory arrest, and death. They lead to failure in the electron transport chain, which, at the mitochondrial level, translates into a blockade of the passage from ADP to ATP. Their inhibitory effect on NADHU-ubiquinone oxidoreductase has been demonstrated in the laboratory and as a consequence of the (ODC) phorbol ester-induced ornithine decarboxylase [60]. Rotenone is specifically classified as an insecticide type II with low toxicity to humans and warm-blooded animals [61]. The selective toxicity of rotenone in insects and fish versus mammals is related to its poor absorption from the gastrointestinal tract of the latter as well as the overall metabolic differences. Rotenone is converted into highly toxic metabolites in insects and fish. On the contrary, it is converted into non-toxic metabolites in mammals. In the motor system, 5-hydroxytriptamine (5HT) can depress GABA-mediated transmission and structures controlling movement [62]. Rotenone and rotenolone showed a great affinity for the membrane receptors of 5HT in our Swiss Target in silico tests, which is indirect evidence of the capacity of these molecules to behave similarly to Ivermectine when it causes helminth muscle paralysis.

## 4. Conclusions

In Latin America, myasis, as a disease, remains somewhat misunderstood. It is excluded from basic epidemiological research and hospital treatments are often lacking [63]. In this framework, traditional ethnomedicine is revealed as a powerful ally to improve the state of health, especially considering cases such as this case study. It can serve as an example that can be replicated by others in the same situation. We have looked for scientific arguments to demonstrate the reason for using extracts of *Lonchocarpus utilis* against south American myasis (tupe). We have provided a summary of the isoflavonoids, prenylated flavonoids, chalcones, and stilbenes that justify the action. We have performed modeling predictions on the affinity of eight chemical components and enzyme targets using Swiss Target prediction software. 

We have concluded that the effects of this extract can be reasonably attributed to an effect of the parasite that causes the disease. Once again, the importance of the plant world in the drug discovery processes must be considered, and a call is made for plant conservation to be used as a source of obtaining added value bioproducts.

## 5. Materials and Methods

### 5.1. Ethnobotanical Study

All the information regarding the study where the data came from is available in Appendix B. It contains references to the voucher specimens of herbarium material collected, the permits and authorizations obtained, and the procedures applied. Table A1 and Table A2 (in Appendix B) summarize the traditional uses given to the species.

### 5.2. Bibliographic Review

A bibliographic study was performed following the Prisma 2009 flow diagram methodology [64]. The databases accessed were Academic Search Complete, Agricola, Agris, Biosis, CABS, Cochrane, Cybertesis, Dialnet, Directory of Open Access Journals, Embase, Espacenet, Google Patents, Google Academics, Medline, PubMed, Science Direct, Scopus, Theseus, and ISI Web of Science. The aim was to find publications on chemical composition and/or activity. The abovementioned ones and the Latin names of the species and synonyms were used as keywords. The selected citations were summarized. Critical reading of this literature allowed us to elaborate on the discussion of the results and main statements.

### 5.3. In Silico Activity Test

Swiss Target prediction software [65,66] was eventually used to investigate the activity in silico to reinforce the principal arguments. Prediction reports were made with the following query molecules:Rotenone;Rotenolone;Deguelin;Tephrosin;3′metoxylupinifolin;4 hydroxylonchocarpin;Lonchocarpene.

## Figures and Tables

**Figure 1 plants-09-00033-f001:**
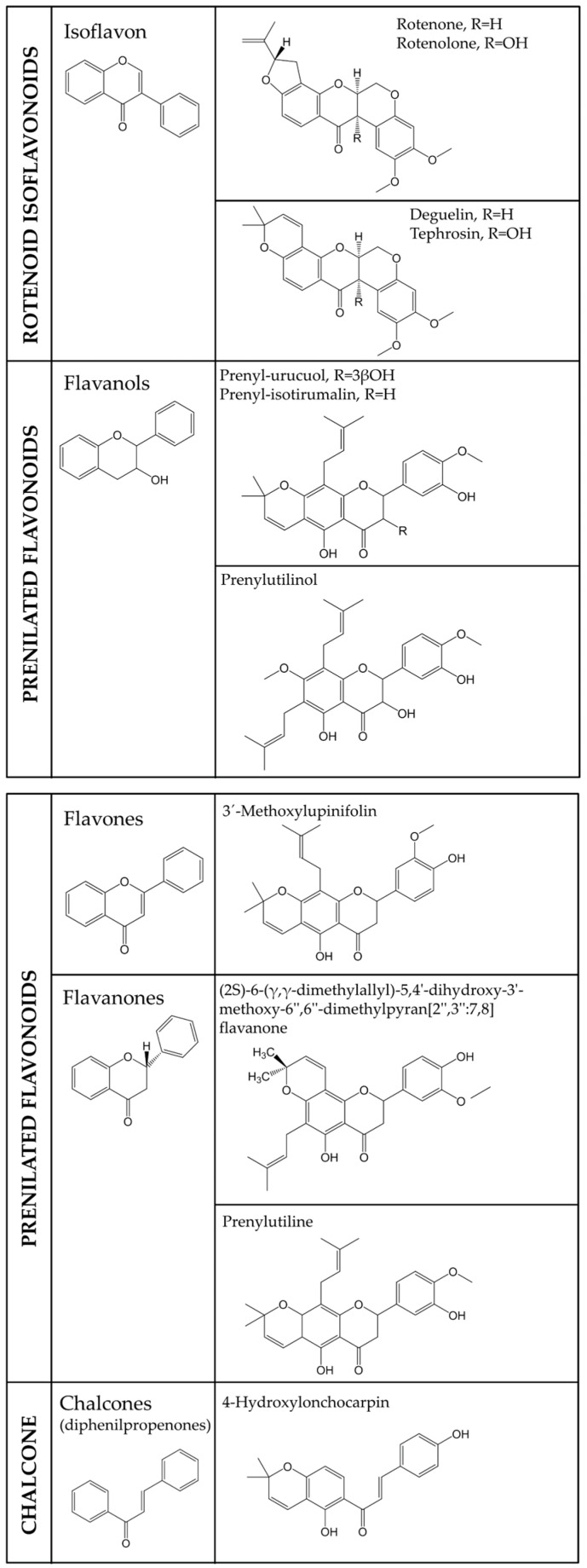
Chemical structures of the main components of *Lonchocarpus utilis.*

**Table 1 plants-09-00033-t001:** Biological activity and applications of some chemical compounds present in *Lonchocarpus utilis.*

Molecule	Tested in	Activity	References
Rotenone	Rat	Inhibition of mitochondrial activity (diminished NADH: ubiquinone oxidoreductase activity)	[41]
	Cell	Inhibition of growth	[31]
	Lehismania	Antilehismaniasic	[42]
	Cell	Antiproliferative	[39]
	Fish	Toxic for fish	[41,43]
	Insect	Insecticide and pesticide	[44]
Rotenolone	Rat	Inhibition of mitochondrial activity (diminished NADH: ubiquinone oxidoreductase activity) (25% less active than rotenone).	[41]
	Cell	Inhibition of growth	[31]
Deguelin		Inhibition of mitochondrial activity (diminished NADH: ubiquinone oxidoreductase activity) (50% less active than rotenone).	[41]
	Cell	Inhibition of growth	[31,45]
	Cell	Antiproliferative	[39]
	Nematode	Nematocide	[46]
		Anti-inflammatory in transplants	[46,47]
	Cell	Potent apoptotic and antiangiogenic	[48,49]
	Cell	Inhibition of progression of tumors such as lung, stomach, prostate, colon, ovary, and pancreas.	[49,50,51,52,53]
	Cell	Inhibition of tumor cell growth and metastasis.	[51,52]
	Cell	Chemical adjuvant against leukemia	[54]
Tephrosin	Rat	Inhibition of mitochondrial activity (diminished NADH: ubiquinone oxidoreductase activity)	[41]
	Cell	Inhibition of growth	[43]
Prenyl-urucuol A	Cell	Cytoprotective activity of neurons in rats (Complete fraction)	[55]
Prenyl-isotirumalin			
Prenylutilinol			
3′-methoxylupinifolin			
Prenylutiline			
(2S)-6-(γ,γ-dimethylallyl)-5,4′-dihydroxy-3′-methoxy-6″,6″-dimethylpyran [2 ″,3″:7,8] flavanone	Cell	Inhibition of growth	[36]
4-hydroxylonchocarpin		Antifungal	[56]
Lonchocarpene	Seedling	Inhibition of growth/development	[57]
4-methoxylonchocarpene	Seedling		
3,5-dimethoxy-4-hydroxy-3-prenyl-*trans*-stilbene	Seedling		

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
