# Peer review of "Promising Potential of *Lonchocarpus utilis* against South American Myasis"

_plants, 2019, doi:10.3390/plants9010033_

Round 1
Reviewer 1 Report
The study “Promising potential of Lonchocarpus utilis against South American Myasis”, is aimed at the validation of a local remedy for myasis, based on bibliographic review of chemical composition and on in silico activity tests.
The aims of the paper is clearly defined. The introduction can be considered updated, and the authors addressed well the subject studied. The discussion is supported by data, being several works of other researchers considered. The paper, according to my perspective deserves publication.
I suggest the authors to improve their paper by adding a short paragraph of conclusions, to point out the results obtained by their study.
Author Response
The study “Promising potential of Lonchocarpus utilis against South American Myasis”, is aimed at the validation of a local remedy for myasis, based on bibliographic review of chemical composition and on in silico activity tests.
The aims of the paper is clearly defined. The introduction can be considered updated, and the authors addressed well the subject studied. The discussion is supported by data, being several works of other researchers considered. The paper, according to my perspective deserves publication.
I suggest the authors to improve their paper by adding a short paragraph of conclusions, to point out the results obtained by their study.
Thank you for your comments, we have included the conclusions paragraph.
Reviewer 2 Report
the manuscript plants-6713 is a well-written manuscript reporting the use of extracts of Lonchocarpus utilis against south American myasis.
In this field, I think that this manuscript is original in the methodology used, but is lack of some information in the introduction. Please add information about the neglected tropical disease, with a particular emphasis on the use of plants and plant extracts.
So far, you have to specify the quantitative amount of substances found in your extract, not only the reported substances by others.
Author Response
Reviewer #2
the manuscript plants-6713 is a well-written manuscript reporting the use of extracts of Lonchocarpus utilis against south American myasis.
In this field, I think that this manuscript is original in the methodology used, but is lack of some information in the introduction. Please add information about the neglected tropical disease, with a particular emphasis on the use of plants and plant extracts.
- Ok. We have added the requested information, and have placed more emphasis on known uses in different countries and ethnicities
So far, you have to specify the quantitative amount of substances found in your extract, not only the reported substances by others.
- As it can be observed in the Material and Methods, we have not collected plant to make extractions and to identify them, because this paper has an ethnobotanical perspective, not a phytochemistry research. That is the main reason of our bibliographic prospections.
Reviewer 3 Report
Authors did a good job of explaining the objective of paper very clearly. In the meantime, I have some questions for the authors:
1) Introduction- Authors need to suggest what other NTDs are and provide a relevant reference in the first couple of paragraphs.
2) Results- I believe that the authors have concentrated too much about work of others. It gets confusing while reading, I will strongly suggest to summarize the work of others and what they really want to accomplish about reading others work and how they have incorporated their findings in their work
3) Overall This paper is written well and has overall merit for publication.
Author Response
Reviewer #3
Authors did a good job of explaining the objective of paper very clearly. In the meantime, I have some questions for the authors:
1) Introduction- Authors need to suggest what other NTDs are and provide a relevant reference in the first couple of paragraphs.
Ok. The requested information has been included in the Introduction block.
2) Results- I believe that the authors have concentrated too much about work of others. It gets confusing while reading, I will strongly suggest to summarize the work of others and what they really want to accomplish about reading others work and how they have incorporated their findings in their work.
Ok. The main reason of reading others work is to get benefit of the knowledge about the activity of the components of the plant extract.
This way, we incorporate all this published information with our ethnobotanical findings (a peculiar use of the plant in a semi non-contacted tribe). Both resources combined with the bioinformatic tools, gives us a valuable data related to new drug discovering. We have included a graphical abstract to illustrate this item.
3) Overall This paper is written well and has overall merit for publication.
Thank you very much.